# Genome-Wide Identification and Spatial Expression Analysis of Histone Modification Gene Families in the Rubber Dandelion *Taraxacum kok-saghyz*

**DOI:** 10.3390/plants11162077

**Published:** 2022-08-09

**Authors:** Francesco Panara, Carlo Fasano, Loredana Lopez, Andrea Porceddu, Paolo Facella, Elio Fantini, Loretta Daddiego, Giorgio Perrella

**Affiliations:** 1Trisaia Research Center, Italian National Agency for New Technologies Energy and Sustainable Economic Development (ENEA), 75026 Rotondella, MT, Italy; 2Department of Agriculture, University of Sassari, Viale Italia, 39a, 07100 Sassari, SS, Italy; 3Department of Biosciences, University of Milan, Via Giovanni Celoria 26, 20133 Milan, MI, Italy

**Keywords:** histone modification, gene expression, *Taraxacum kok-saghyz*, natural rubber

## Abstract

*Taraxacum kok-saghyz* (*Tks*), also known as the Russian dandelion, is a recognized alternative source of natural rubber quite comparable, for quality and use, to the one obtained from the so-called rubber tree, *Hevea brasiliensis*. In addition to that, *Tks* roots produce several other compounds, including inulin, whose use in pharmaceutical and dietary products is quite extensive. Histone-modifying genes (HMGs) catalyze a series of post-translational modifications that affect chromatin organization and conformation, which, in turn, regulate many downstream processes, including gene expression. In this study, we present the first analysis of HMGs in *Tks*. Altogether, we identified 154 putative *Tks* homologs: 60 HMTs, 34 HDMs, 42 HATs, and 18 HDACs. Interestingly, whilst most of the classes showed similar numbers in other plant species, including *M. truncatula* and *A. thaliana*, HATs and HMT-PRMTs were indeed more abundant in *Tks*. Composition and structure analysis of *Tks* HMG proteins showed, for some classes, the presence of novel domains, suggesting a divergence from the canonical HMG model. The analysis of publicly available transcriptome datasets, combined with spatial expression of different developmental tissues, allowed us to identify several HMGs with a putative role in metabolite biosynthesis. Overall, our work describes HMG genomic organization and sets the premises for the functional characterization of epigenetic modifications in rubber-producing plants.

## 1. Introduction

The genus *Taraxacum Wigg*. (dandelion) is included in the large family of Asteraceae (Compositae), which counts more than 2800 species [1]. Most of these species of dandelions live in the temperate zones of the northern hemisphere, although they are native to Eurasia [2]. Among all the dandelions species, the Russian dandelion (*Taraxacum kok-saghyz*, *Tks*) can produce from its roots natural rubber (NR) of excellent quality, quite comparable to that obtained from *Hevea brasiliensis*, the so-called rubber tree [3]. In *Tks*, the length of the rubber polymer is greater than that of Hevea, with a molecular weight of approximately 2180 kDa, but its productivity per hectare is lower [4]. Nevertheless, *Tks* mature roots can contain up to 5% of rubber and 20% of inulin in dry weight in wild plants and, therefore, this dandelion is considered as a valid alternative natural rubber source. *Tks* can extensively grow both in cold and temperate areas, presents a short life cycle, and can be easily harvested [5]. Moreover, *Tks* is a species suitable for genetic manipulation and it could be used as a model plant to investigate the molecular pathways that regulate NR biosynthesis [6,7].

NR is a polymer composed of cis-1,4-polyisoprene and it represents a very important raw material, used to produce more than 50,000 both industrial and medical products [8].

The *Tks* genome has been recently published, showing that this dandelion (a diploid species 2 *n* = 16) has a genome size of 1.29 Gb, comprising 46,731 predicted protein-coding genes [9].

*Tks* presents tissues specialized in the production of NR, called laticifers [3]. Laticifers contain latex in specific structures, the rubber particles, that are spherical organelles surrounded by a lipid monolayer membrane [8]. Several proteins are bound to the membrane of the rubber particles and some of them play an important role in rubber biosynthesis. In the roots of Taraxacum spp., there are also present in significant quantities the carbohydrate inulin and other metabolites that are relevant in various applications [2,10,11,12,13].

In plants, chromatin is comprised in nucleosomes that are constituted by two copies of each histone protein (H2A, H2B, H3, and H4) wrapped around ~146 base pairs of DNA [14]. Together with canonical histone proteins, nucleosomes can also contain histone variants that are usually integrated in specific chromatin regions and in response to external stimuli [15]. Histones are composed of N-terminal tails that are enriched in basic amino acids that are subjected to different post-translational modifications (PTMs) [16]. The simultaneous presence of different modifications defines the so-called histone code that is superimposed over the genetic code to regulate most cellular mechanisms [17]. Histone H3 and H4 tails can indeed be methylated, acetylated, and phosphorylated at different levels (mono-di-and tri). Interestingly, different lysine and arginine residues can be subjected to different modifications that ultimately work as positive or negative regulators of gene expression [18]. Histone methylation is catalyzed by histone methyltransferases (HMTs) and can occur primarily at H3 Lys4 (K4), Lys9 (K9), Lys27 (K27), and Lys36 (K36) [19]. HMTs comprise mostly of the SET domain group (SDG) enzymes that are divided in seven different classes: (1) E(Z) (enhancer of zeste) homologs, (2) ASH1 (absent, small, or homeotic discs 1) groups (ASH1 homologs (ASHH) and ASH1-related proteins (ASHR)), (3) TRX (Trithorax) groups (TRX homologs and TRX-related proteins), (4) SET and PhD domain, (5) SU(VAR)3–9 groups (together with SU(VAR)3–9 homologs (SUVH) and SU(VAR)3–9-related proteins (SUVR)), (6) truncated SET domain class and (7) Rubisco large (RBLSMT) and small (RBSSMT) subunit methyltransferases SET-related class [20,21,22]. In addition to that, plants also do present a second HMT family that includes the protein arginine methyl-transferases that contain the PRMA domain (PRMTs) [23].

While histone methylation is catalyzed by HMTs’, histone demethylases (HDMs) are responsible for its removal [24]. In plants, there are two main classes of HDM enzymes that are distinguished through their mechanism of action. Indeed, the KDM1/LSD1-likes (HDMAs) operate through amine oxidation to remove methylation marks, while JmjC domain-containing (JMJs) demethylases use hydroxylation [25,26,27]. At the same time, cofactors and the substrates change based on the type of HDMs. HDMAs only demethylate residues that have been mono- or dimethylated by a flavin adenine dinucleotide (FAD)-dependent reaction. Instead, JMJs demethylate lysines independently on the type of methylation and use Fe(II) and α-ketoglutarate (αKG) as cofactors [28].

Histone acetyltransferases (HATs) and deacetylases (HDACs) catalyze the addition and the removal of the acetyl-CoA groups over the histone tails, respectively [29]. This modification is usually correlated with changes in gene expression as well as chromosome condensation. In Arabidopsis, HATs are divided in four groups: GNAT and MYST families (HAGs and HAMs, respectively), CBP (HACs), and the TAFII250 (HAFs) [30]. HAGs are then classified in three main classes: GCN5 or GCN5-likes, ELP3, and HAT1. MYSTs, instead, contain mostly HAM members [31]. HDACs in plants are grouped in three main groups: RPD3/HDA1 superfamily, similar to the yeast large RPD3 complex (HDAs), the sirtuins (SIR2/SRTs), and the HD2/HDT enzymes, whose class appears to be specific in plants [32,33]. Both HATs and HDACs have been largely characterized in plants. Indeed, their function is involved in different developmental processes and transitions, including germination, cell differentiation, and leaf and floral organogenesis [34,35,36,37]. Furthermore, their role has been linked with changes in gene expression upon different abiotic and biotic stress conditions, such as salinity, light, temperature, and immuno-responses [38,39,40,41,42].

Here, we report for the first time an extensive in silico analysis and identification of the histone-modifying genes (HMGs) in *Tks*. Using publicly available genome and transcriptome data, we analyzed the gene and protein structure of the identified HMGs and monitored their expression in various *Tks* tissues, including latex. In addition, we measured transcript levels in *Tks* young and adult leaves and roots and identified those per each class of enzymes that appeared to be abundant in relevant tissues.

## 2. Results

### 2.1. HMG Genes Identification

In the *Tks* genome [9]***,*** we identified a total of 154 HMGs: 60 HMTs, 34 HDMs, 42 HATs, and 18 HDACs. We compared the total number with a close relative of *Tks*, *Lactuca sativa* (Ls) and with the reference plants *Arabidopsis thaliana* (At) [43] and *Medicago truncatula* (Mt) [44]. While some classes were similar in numbers, others showed divergent expansion among different species. PRMTs were indeed more abundant in *Tks*, indicating an expansion of this class of proteins similarly to *Litchi chinensis* [45]. Altogether, HAGs were more abundant in both *Tks* and *Ls* compared to *At*, confirming the possibility for this group to expand divergently in different taxa (Table 1). The complete list of *Tks* HMGs is reported in Appendix A.

Gene duplication events contributed to the high number of HMGs in the *Tks* genome. Synteny analysis on the *Tks* genome assembly was not effective for the identification of segmentally duplicated HMGs, while five couples of tandemly duplicated genes were identified: *Tks*HDA5/*Tks*HDA6, *Tks*HAG30/*Tks*HAG31, *Tks*HAG36/*Tks*HAG37, *Tks*JMJ11/*Tks*JMJ12, and *Tks*SDG44/GWHPAAAA005121. In the last pair, GWHPAAAA005121 was not classified as an HMG due to the absence of a representative domain. We identified 32 additional couples of putative duplicated HMGs based on phylogenetic analysis. The Ka/Ks ratio was calculated for all the gene pairs in order to estimate the occurring evolutionary dynamics (Appendix A). Most of the couples showed Ka/Ks <1, suggesting purifying or stabilizing selection. Only the *Tks*HAG13/14 pair with a Ka/Ks ratio of 1.54 and *Tks*PRMT7/8 with 1.05 showed values compatible with positive and neutral selection.

#### 2.1.1. HMTs

Among HMTs, 46 were identified belonging to SDG and 14 to the PRMT group. SDGs are similar in number to *At* and the same amount in *Tks* and *Ls* (Table 1). SDGs are divided in seven classes [22,46]. Four *Tks*SDGs cluster with Class I, E(Z)-like SDGs (Figure 1A) and show the expected SANT-CXC-SET domains except for *Tks*SDG8 lacking the SANT *domain* (Figure 2).

Five clusters with Class II, ASH1-like. *Tks*SDG29, *Tks*SDG16, and *Tks*SDG19 have the same domain composition of corresponding *At* proteins. *Tks*SDG15 encodes for a short (52aa) and a probably aberrant protein where only the SET domain was identified. *Tks*SDG37 lacks the PostSET domain compared to AtSDG26. Five proteins cluster with Class III, TRX-like. *Tks*SDG41 differs from AtSDG14 as the first PHD module is substituted by a SAND domain that is often associated to PHDs and could contribute to their function. *Tks*SDG45 showed an additional TUDOR domain at the N-terminal. The TUDOR domain was previously identified in *D. melanogaster* TUDOR proteins and its function is unknown. Several human JMJ/KDM4 proteins harbour a TUDOR domain at the C-terminal [44]. *Tks*SDG4 clusters with *At*SDG25 but differs in domain composition and sequence length: GYF-SET and 1057aa the former, and SET-PostSET 1424aa the latter. This was previously observed in *S. lycopersicum* (*Sl*), where *At*SDG25 and *Sl*SDG20 showed similar differences [45]. Two Class IV SDGs exist in both *Tks* and *At*. Six *Tks* proteins cluster with class V subclass I and six with class V subclass II with a similar domain composition to *At* proteins. Eighteen SDGs can be classified as belonging to class VI/VII and show an interrupted SET domain. *Tks*SDG40 is a short protein of 158aa, probably aberrant, and the presence of the SET domain was not confirmed by SMART analysis.

*Tks*SDG5 and *Tks*SDG25 encode for putative long proteins with an additional/unexpected domain composition, probably resulting from exon gain or, more likely, a defect in genome assembly/annotation.

As shown in Appendix A, three PRMTs form a separate cluster: *Tks*PRMT2, *Tks*PRMT7, and *Tks*PRMT8 and contains two PRMT domains (Figure 2). The PRMT5 domain was not confirmed for *Tks*PRMT9 and *Tks*PRMT11 by SMART analysis. *Tks*PRMT6 shows four C2H2 modules at the N-terminal similarly to HMGs belonging to other groups such as *At*SDG6/*Tks*SDG24 (Figure 2).

#### 2.1.2. HDMs

In *Tks*, we identified 27 proteins containing the JmjC domain (PF02373) (Table 1). Despite that nine harbour the JmjC domain alone and could be classified as JMJ-only, only *Tks*JMJ25 and *Tks*JMJ27 clustered with *At* JMJ-only proteins (Figure 1B and Figure 2). The other seven are closer to other JMJ groups but underwent the loss of representative domains. In addition, only four out of fourteen proteins show the presence of the RING domain typical of the KDM3 group (Figure 2). KDM4 JMJs present N-terminal JmjN (PF02375) and JmjC, and C-terminal C5HC2 (PF02928) (subgroup I) or C2H2 (PF00096) (subgroup II). In *Tks*, two proteins belong to subgroup I and three to subgroup II. Four proteins cluster with KDM5 JMJs and show the expected domain composition (Figure 1B and Figure 2).

In addition, among *Tks*JMJs, several proteins revealed additional C-terminal domains (*Tks*JMJ19, *Tks*JMJ3, *Tks*JMJ16, and *Tks*JMJ26) or exceptionally long introns (*Tks*JMJ7) that likely derive from errors in genome annotation (Figure 2). In the JMJD6 group, *Tks*JMJ15 has the expected F-Box domain while *Tks*JMJ5, although clustering with AtJMJ21, retains only the JmjC domain (Figure 1B and Figure 2).

Seven HDMAs were identified by our analysis (Table 1). *Tks*HDMA1, *Tks*HDMA7, and *Tks*HDMA4 are grouped with *At*HDMA4, 2, and 1, respectively (Appendix A). The above-mentioned proteins and *Tks*HDMA6 show the presence of SWIRM-NADB8-AOD domains (Figure 2). *Tks*HDMA2, *Tks*HDMA3, and *Tks*HDMA5 form a subgroup harbouring a SANT domain (Appendix A and Figure 2).

#### 2.1.3. HATs

We identified 42 HATs in *Tks*. Most of them, 38, belong to the HAG group (Table 1). An expansion of HAGs compared to *At* was already observed in other species analysed so far such as *Medicago truncatula* [44], *Malus domestica* [26], *Solanum lycopersicum* [47], *Citrus sinensis* [48]***,*** and *Vitis vinifera* [49].

GCN5-type HAGs, *Tks*HAG36, and *Tks*HAG37 have a C-terminal Bromodomain similar to *At*HAG1. Three additional proteins clustering with *At*HAG1 are shorter and probably underwent loss of the Bromodomain: *Tks*HAG8 193 aa, *Tks*HAG33 250 aa, and *Tks*HAG34 192 aa (Figure 3A and Figure 4 and Appendix A).

*Tks*HAG16 clusters with *At*HAG2 and harbours the Hat1 domain. *Tks*HAG10 clusters with *At*HAG3 and harbours the ELP3 domain. Two additional proteins clustering with *At*HAG3, *Tks*HAG27, and *Tks*HAG6 show the AT1 domain alone (Figure 3A and Figure 4). Furthermore, other AT1 domain-containing proteins can be observed.

A first group of seven proteins forms a definite cluster that is characterized by the presence of Jas and PHD domains except for two shorter proteins, probably derived from the others, that lost specific domains: *Tks*HAG21 and *Tks*HAG25. In *Tks*HAG32, the AT1 domain was not identified as below the threshold for SMART analysis (Figure 3A and Figure 4).

Ten proteins are characterized by the presence of an additional FR47 domain at the C-terminal. Four of them, *Tks*HAG5, *Tks*HAG11, *Tks*HAG17, and *Tks*HAG19 form a definite cluster. Instead, a second cluster includes *Tks*HAG14, *Tks*HAG15, *Tks*HAG23, *Tks*HAG29, and two additional proteins: *Tks*HAG20 where no domain was identified by SMART, and *Tks*HAG24, where the FR47 domain is missing and there is an N-terminal AAK domain. Interestingly, *Tks*HAG13, containing both AT1-FR47, do not cluster with the other FR47 domain-containing proteins (Figure 3A and Figure 4).

Two proteins, *Tks*HAG9 and *Tks*HAG26 show two C-terminal BRCT domains and form a cluster with *Tks*HAG30 and *Tks*HAG31 that are shorter and without a BRCT domain. *Tks*HAG12 harbours AT1 and C2 domains. All other *Tks*HAG proteins show the AT1 domain alone (Figure 3A and Figure 4).

One MYST acetyltransferase was identified: *Tks*HAM1 with Chromo-C2H2-MYST-Syja_N domain composition (Appendix A).

Two HACs identified in *Tks* have a TAZ-PHD-KAT11-ZZ-ZZ-TAZ domain composition, a third HAC, *Tks*HAC2, lost the N-terminal TAZ domain and clusters with *At*HAC2 (Appendix A). No proteins showing the HAF domain were identified by our analysis.

#### 2.1.4. HDACs

We identified 18 histone deacetylases (Table 1). Nine are HDAs, three belong to class I, RPD3-like, five to class II, HDAC1-like, and one to class III, HDAC11-like. In class I, *Tks*HDA9, clustering with *At*HDA19, is a long protein (1634 aa) showing six additional N-terminal TPR domains probably resulting from exon gain or a defect in genome assembly/annotation. In the class II, *Tks*HDA7 similarly to *Mt*HDA8 [44] harbours an N-terminal Znf_RBZ domain (Figure 3B and Appendix A).

Four proteins are SRTs containing the SIR2 (PF02146) domain. Interestingly, with the exception of *Tks*SRT4, all other SRTs have a double SIR2 domain (Appendix A).

Five *Tks*HDTs were identified by BLASTp analysis (Table 1). The most conserved are *Tks*HDT1 and *Tks*HDT2 if compared with *At*HDTs. *Tks*HDT1 showed a C2H2 domain. In *Tks*HDT5, C2H2 modules are six (Appendix A). In plants, HD2-Types are related to cis–trans isomerases found in insects and yeast [30]. Similarly, their function as HDACs remains debatable.

### 2.2. In Silico Expression Analysis of HMG Genes in Different Organs and Developmental Stages

To shed light on the spatial expression of the HMGs in *Tks*, we analyzed publicly available datasets [9], based on the following tissues/organs: leaves, stems, the main root, and the lateral root in both young and mature stages (eight samples); latex and three additional samples from reproductive organs: flower, peduncle, and seeds. Expression patterns of HMGs were evaluated for each gene as log_2_ fold change (Fc) compared to the average among samples. Hierarchical clustering was used to assess groups with similar expression patterns.

#### 2.2.1. HMTs

Among SDGs, a first group presents a peak of expression in young/mature leaves compared to other organs. Within this group, *TksSDG28* shows an additional peak in the latex. A second large group includes genes with higher expression in reproductive organs, mainly flowers and seeds. In group 3 we identified genes with higher expression in young organs and flowers. Group 4 shows higher expression in stem and roots with lower expression in the latex. The last group harbours the most expressed genes, showing an expression pattern primarily in mature roots (Figure 5).

*Tks*PRMTs exhibit less modulated expression patterns. Three groups can be identified: the first with higher expression in latex and reproductive organs; the second with a peak in flowers; and the third group with higher expression in stem and roots. In the first group, *TksPRMT7* is the one showing the most abundant peak of expression in latex (Figure 5).

#### 2.2.2. HDMs

Among *Tks*JMJs, a first group includes genes with higher expressions in reproductive organs. In addition to that, *TksJMJ22* presents a relevant peak also in the latex, indicating a strong modulation compared to whole roots. A second group includes genes with a higher expression in stem-root and flower, and little expression in latex and leaf. The third group shows a higher expression in stem-root but, differently from most JMJs, not in flower. This group includes JMJs with high levels of expression. The fourth group shows a relevant expression in stem and flower (Figure 5).

All HDMAs exhibit higher expression in stem, flower, and seeds, with two groups that can be distinguished by high and low expression in mature root and latex (Figure 5).

#### 2.2.3. HATs

Four groups were identified among *Tks*HAGs: a first one with peaks in the roots (mainly in mature roots); a second group predominantly in reproductive organs; a third with the same pattern with addition of leaves; and interestingly, the fourth group is made of highly expressed genes showing a peak in the latex similarly to the single HAM gene detected by our analysis. HACs are less expressed in leaves and latex (Figure 6).

#### 2.2.4. HDACs

*Tks*HDAs can be divided into three groups: a first one with peaks of expression in leaves and peduncle; a second group with lower levels in the roots; and a third group more relevant in young and reproductive organs. *TksHDA8* shows a peculiar pattern with peaks in the main root and latex. *Tks*SRTs were present in reproductive organs with *TksSRT4* and *TksSRT2* having an additional peak in mature leaf and in the latex, respectively. HDTs, instead, show peaks of expression in young leaf, stem, lateral root, and in the mature stem with the exception of *TksHDT5*, which shows peaks in reproductive organs.

### 2.3. HMGs Expression

Using the absolute quantification method, we measured the expression of 13 different *Tks*HMGs belonging to four families in four different tissues: developing leaves (DL), fully developed leaves (FL), root tips (RT), and mature roots (R). Figure 7 shows the spatial gene expression pattern of four HMTs (*TksSDG3, TksSDG4, TksSDG24,* and *TksPRMT9*), five HDMs (*TksJMJ14, TksJMJ23, TksJMJ25, TksHDMA4*, and *TksHDMA6*), one HAT (*TksHAG10*), and three HDACs (*TksHDA7*, *TksHDA8*, and *TksHDA9*). Five genes (*TksHDMA4, TksHDA9, TksHDMA6, TksJMJ23*, and *TksHAG10*) have at least one tissue expression level higher than 1 femtogram (fg)/μL (Figure 7a). The most expressed gene is *TksHDMA4*, with an average tissue quantification of 6 fg/μL but no significative modulation between the different tissues. Among the remaining eight genes, *TksSDG3* and *TksJMJ25* expressions never exceed 0.1 fg/μL (Figure 7b). With an average tissue quantification of 0.05 and 0.01 fg/μL, respectively, they are the lowest expressed HMG genes. Overall, there is no difference of global HMG gene expression between the four different tissues analyzed, with an average expression of 1.38 and 1.29 fg/μL in fully developed leaves (FL) and the root (R), respectively. *TksJMJ25* and *TksHDA8* expression increased significantly, 319 and 174%, respectively, in FL in comparison to DL. *TksHAG10*, *TksSDG24*, and *TksPMRT9* are up-regulated in a less strong way. The remaining eight genes confirm an FL up-regulation with respect to DL; however, below a 50% change. *T*ks*SDG4*, *TksHDA8*, and *TksSDG3* expression increased clearly with a 276, 249, and 104%, respectively, in R compared to RT. *TksHDA7* and *TksHAG10* are up-regulated by 85 and 56%. As for leaves, all eight remaining genes are up-regulated in R tissue in comparison to RT with a percentage below 50%.

## 3. Discussion

### 3.1. HMG Members in Taraxacum Kok-Saghyz

Natural rubber represents a biopolymer of major importance due to its wide properties that cannot be found in synthetic material, including resilience, elasticity, and abrasion [50]. The demands to develop new sources of natural rubbers have quickly increased over the years due to the constant reductions of petroleum-based materials and the efforts towards renewables [3]. Currently, the main resource for commercial natural rubber, *Hevea brasiliensis*, has generated major concerns because of its high sensitivity towards pathogen infections [51]. Furthermore, latex can induce several allergic reactions due to their protein content [52]. As an alternative to *H. Brasiliensis*, two plant species are so far known to produce rubber in similar amounts and a high molecular weight: *Parthenium argentatum* Gray, also known as guayule, and *Taraxacum kok-saghyz*, the Russian dandelion [53]. *Tks* latex, for instance, is broadly investigated for non-medical applications such as tire production [54].

In *Tks*, rubber biosynthesis occurs through different pathways that are responsible for NR chain elongation, small rubber particles (SRPPs), and rubber elongation factors, respectively [7]. Therefore, to improve rubber quality and quantity, it becomes paramount to functionally characterize the genes involved in those pathways.

Histone modifications are well established to control gene expression [15]. Thus, they represent a promising strategy to manipulate different signalling pathways, unlike the gene-by-gene approach [55].

In this study, we presented for the first time a comprehensive analysis of the HMGs in *Tks*, identified by the presence of representative domains recognized through their hidden Markov profiles. HMGs sequences and/or composition were compared with the model species *A. thaliana* and *M. truncatula*, as well as the close relative *L. sativa*.

Overall, the identification recorded an increase in the number of HMG genes in *Tks* (154) compared to *A. thaliana* (102), which is similar to what is observed in *L. sativa* and *M. truncatula*. Due to limitations in the assembly, it was not possible to assess gene duplication.

A closer investigation revealed that such an increase did not occur on all the HMG classes, but it was indeed specific. Thus, while HDMs were similar in numbers (34 in *Tks* vs. 24 in Arabidopsis), HATs were predominantly higher in the former (42) than in the latter (12). This is not necessarily surprising, as an expansion of HAGs was already revealed in other crops when compared to Arabidopsis (Table 1) [44].

Among the HMTs family, analyses on *Tks* indicated the presence of 46 genes encoding for SDGs, a similar number to those of lettuce and Arabidopsis. Instead, the PRMTs were doubled up in *Tks* compared to the other species. Overall, the *Tks* SDGs showed a similar structure to those already annotated with SANT-CXC-SET domains except for *Tks*SGD8 that showed a shorter structure (Figure 2). SANT domains are essential to ensure histone substrates for the SGD enzymatic activity [56], while the CXC allows binding to RNA molecules as the SET represents the signature domain to cluster the SDGs [57,58]. *Tks*SDG5 appears to be much longer than the other SDGs, perhaps due to fusion between two different proteins (Figure 2). The presence of a bag6-A domain could implicate a divergent function among the other SDGs, including a possible interaction with heat shock proteins [59]. A functional study is therefore required to address this question.

Deeper investigation on the PRMTs indicated that *Tks*PRMT2, 7, and 8 contained two PRMT domains whilst the majority had only one. Structural analyses combined with domain deletion could retrieve more information on that regard.

*Tks* HDMs were classified in JMJs and HDMAs. Interestingly, many JMJ-only clustered with the KDM3 groups although they lacked the additional RING domain (Figure 5). However, they all presented a very extensive introns and exons structure that is typical of the KDM3s [60].

Over 90% of *Tks* HATs belonged to the HAG group. Similar to previous reports [48], the majority of the members contained only the AT1 protein domain (Figure 4). Whilst *Tks*HAG36 and 37 were identified as GCN5 likes due to the presence of the bromodomain, others included Jas and PhD domains or the FR47, or the BRCT domain in combination with AT1. This appears to be quite divergent from the canonical HAGs structure, suggesting a novel class of HMG HATs in *Tks*. Indeed, Jas domains are present in JASMONATE ZIM-domain (JAZ) proteins that are repressors of jasmonate (JA) signalling [61]. FR47 domain resembles the C-terminal region of the Drosophila melanogaster hypothetical protein FR47. Interestingly, a member of the HATs in *Plasmodium falciparum* also contains that domain, suggesting a class of acetyltransferases targeting nonhistone proteins [62].

Our approach identified 18 HDACs in *Tks*. The three classes showed numbers comparable to Arabidopsis (Appendix A). Interestingly, *Tks*HDA9, which shows similarities with *At*HDA19, appears to have additional domains. *At*HDA19 is a major regulator of development and growth [41,63]. Whether *Tks*HDA9 has developed similar functions remains to be assessed.

### 3.2. In Silico and Spatial Analysis of HMGs in Tks Tissues Identified Putative Regulators of NR Biosynthesis

Histone modifications are well known to regulate important functions in plants [15]***,*** but very little is known about their expression patterns and behaviour in *Tks*. To unveil new roles for the HMGs, we took advantage of publicly available datasets in *Tks* and measured their mRNA levels as fold-change normalized to the average amounts. Among the different tissues analyzed, latex represented the most interesting as it is directly connected to natural rubber production. Indeed, the biosynthesis of NR occurs in the latex of laticifers, organized in rubber particles [64]. Within the SDG class VI, *TksSDG28* presented a distinct expression in latex (Figure 5). Instead, *TksSDG3* and *4* showed an abundant level of transcripts in young leaves and roots. Interestingly, *TksSDG4* was more abundant in roots, whilst *TksSDG3* showed similar levels in both tissues. The Arabidopsis *Tks*SDG3 homolog, SWINGER, is part of a large protein complex that includes VRN2 (VERNALIZATION 2) and VIN3 (VERNALIZATION INSENSITIVE 3), together with FERTILIZATION INDEPENDENT ENDOSPERM (FIE) and CURLY LEAF (CLF). They are responsible for establishing FLC (FLOWERING LOCUS C) repression during vernalization [65]. Notably, the *Tks*SDG4 homolog in Arabidopsis is ATXR7 that instead promotes H3K4me3 on the FLC locus, hence its transcriptional activation [66]. A similar function is associated with *At*SDG8, a homolog of *Tks*SDG24 [67]. Whether their function is indeed conserved in *Tks* and is based on tissue specificity remains to be assessed.

Heatmaps for *Tks* PRMTs did not show major modulations among tissues, with the exception for *TksPRMT9* and *Tks*P*RMT11* that portrayed an elevated expression in young leaves (Figure 5). Real time qPCR analyses of some of the indicated genes revealed a tendency towards root samples. Interestingly, *TksSDG4* was expressed primarily in mature roots and *TksPMRT9* showed major peaks in fully developed leaves as well as roots, making them interesting targets for a functional approach (Figure 7b).

In silico expression of the histone demethylases revealed a predominance on the reproductive tissues (Figure 5). However, *TksJMJ14* showed a preferential expression in roots. Our qRT-PCR data showed instead that levels were different between root tips and adult roots, suggesting a modulation associated with plant development. Furthermore, it is important to highlight that, while mature roots contain laticifers already producing rubber, in the root tips the laticifers are still developing. Therefore, our qPCR analysis presents dual significance: first, it identifies putative HMGs involved in regulating the expression of genes related to rubber or laticifers production; second, the comparison between the expression in leaves and roots is pivotal to distinguish between overall growth and rubber-specific mechanisms. The analysis of other two JMJs *TksJMJ23*, a homolog of Arabidopsis ELF6 [67] and *TksJMJ25*, confirmed in silico results as none of them were modulated but instead they showed similar expression levels among the analyzed tissues (Figure 7a,b). A similar pattern was observed for the HDMAs, where only *TksHDMA4* turned out to be the most abundant among the analyzed genes. Interestingly, the heatmaps showed a major expression in the latex tissue.

Heatmaps for histone acetyltransferases indicated only a few HATs were expressed in root tissues, whilst the group 3 was majorly present in young and mature leaves (Figure 6). We focused our attention on *TksHAG10* that indeed showed to be mostly abundant in roots (Figure 7a). Interestingly, our qPCR data also showed a peak in fully developed leaves that was not observed in the datasets analysis. However, the plant material used in the datasets experiment might not fully correspond to the one obtained for expression analysis.

Among the histone deacetylases, *TksHDA9* was the most expressed within the family (Figure 7a). Interestingly, *TksHDA9* revealed a drop in expression between leaves and roots, while *TksHDA8* was instead more abundant in mature roots. It will be interesting to assess whether *TksHDA9* action is counteracted by histone acetyltransferases GCN5-likes.

## 4. Materials and Methods

### 4.1. Plant Material

A *Tks* plant belonging to the W6-35166 population obtained from USDA-ARS (Regional Plant Introduction Station, 295 CLARK HALL, WSU Pullman, WA 99164, Washington State University, USA) was grown and clonally propagated from root cuttings. Fully developed clones have been cultivated in trays filled with topsoil under controlled conditions (12/12 light cycle, 20 °C) for three weeks, then transplanted in 85 × 39 × 34 cm rectangular pots, 8 plants each. Potted plants were cultivated from January to November 2021 under near natural conditions for photoperiod, temperature, humidity, and exposure to atmospheric events, in ENEA Trisaia Research Center (40°09′47.4″ N 16°38′00.1″ E, Italy). Plants were supplemented with fertilisers, treated against pests and fungal pathogens, and shaded during summer. Four different tissues have been harvested and snap-frozen in liquid nitrogen: developing leaves, fully developed leaves, root tips, and mature roots.

### 4.2. In Silico Identification and Analysis of HMG Loci

To identify HMGs loci, the hidden Markov profiles of each gene family were used as a query input for the HMM software against the protein subject datasets of Taraxacum kok-saghyz and Lactuca sativa [9]. Since for the HDT family no PFAM domain is available, we used Arabidopsis HDTs as query input for BLASTp searches against the protein datasets. The identified loci were analysed with the SMART software to characterise the domain composition [68,69]. The cases pointing to two different domains for the same protein region were resolved based on the domain that was associated to the smaller E-value. The intron/exon structures were graphically represented with GSDS 2.0 [70].

### 4.3. Phylogenetic Analysis

The HM protein families identified in this work from Taraxacum kok-saghyz genome annotation database [9] along with HM proteins from Arabidopsis were aligned using ClustalW program in CLC Genomics Workbench version 9.5.2 (Qiagen, Hilden, Germany). A phylogenetic tree was then constructed using a neighbor-joining algorithm and Kimura protein substitution model. Reliability of the internal branching was obtained by a bootstrap test of 1000 replicates.

Collinearity analysis was performed using MCSCANX [71]. In brief, Taraxacum proteins were searched for sequence homology using BLASTp [72] according to McScanX default settings. Collinear blocks were identified by McSCANX with default settings and sequence duplication were classified with the duplicate gene classifier tool integrated in the McSCANX suite. Protein of duplicated HMG were aligned with ClustalO [73] and codon-based alignments were obtained with PRANK software [74]. The synonymous and non-synonymous substitution patterns were calculated with PAML [75] using YN00 procedure [76]. 

### 4.4. Transcriptome Analysis

RNA reads included in project PRJCA000437 were downloaded from the National Genomics Data Center (NGDC), part of the China National Center for Bioinformation (CNCB) and quality checked using Trimmomatic [77] according to Corchete et al., (2020) [78]. The reads were then aligned to the *Tks* indexed genome using STAR [79] with default settings. HTSeq-count script [80] was used to determine raw counts for each feature and experiment. The raw count matrix was normalised using edgeR and TMM method [78,81].

### 4.5. RNA Extraction, cDNA Synthesis, and Gene Expression Analysis

Total RNA from the above-mentioned plant material was extracted using innuPREP Plant RNA Kit (Analytik, Jena, Germany) according to the manufacturer’s protocol. RNA quality and concentration were estimated by Nanodrop™ 1000 Spectrophotometer (Thermo Fisher Scientific Waltham, MA, USA). A total of 1 µg of total RNA was used for cDNA synthesis using SuperScript IV kit (Thermofisher Waltham, MA, USA) with oligo(dT)_20_ as primers in a 20 μL final volume. Primer pairs for SYBR-Green based qPCR have been designed for each candidate gene using Geneious Prime 2022.1.1 (https://www.geneious.com). Each primer has been subjected to off-target analysis using local BLAST-2.11.0+ [72,82,83] against the Taraxacum kok-saghyz genome annotation [9] downloaded from https://ngdc.cncb.ac.cn/gwh/. Pairs with both forward and reverse primers matching an off-target transcript were rejected. Primer sequences are reported in Appendix A. qPCR was performed using an ABI Prism 7900HT instrument (Applied Biosystems, Waltham, MA, USA) and Platinum SYBR Green qPCR SuperMix-UDG with ROX (Thermofisher, Waltham, MA, USA) following manufacturer’s instructions. Reactions were performed in two technical replicates on three biological replicates. The following cycling conditions were used for quantitative PCR: 5 min at 95 °C followed by 45 cycles at 95 °C for 15 s and at 58 °C for 60 s. Melting curve analysis from 60 to 90 °C was performed to monitor the specificity of the amplification. mRNA levels of each genes analysed were calculated using absolute quantification based on the standard curve method [84] as reported by D’Amelia et al., (2014) [85] expect for standard curves obtained by purified and quantified conventional PCR normalised to a concentration of 33 pg/μL and 10-fold serial dilutions (ranging from 10^2^ to 10^8^). Expression data were analysed using Tukey’s pairwise test.

## Figures and Tables

**Figure 1 plants-11-02077-f001:**
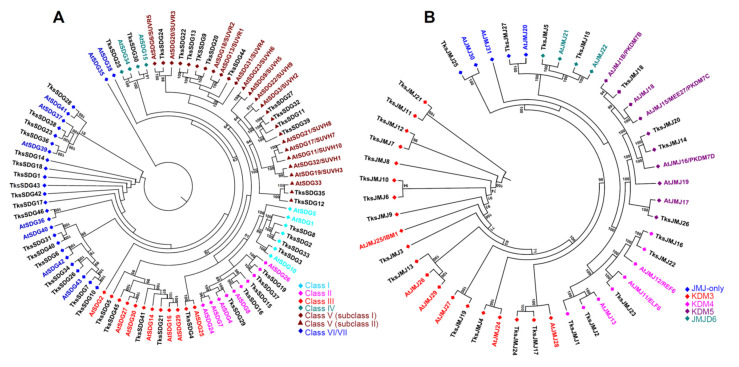
Phylogenetic tree of SDG (**A**) and JMJ (**B**) proteins of *Taraxacum kok-saghyz* and *Arabidopsis thaliana* (in colour according to their respective classes). The numbers near the tree branches represent bootstrap values.

**Figure 2 plants-11-02077-f002:**
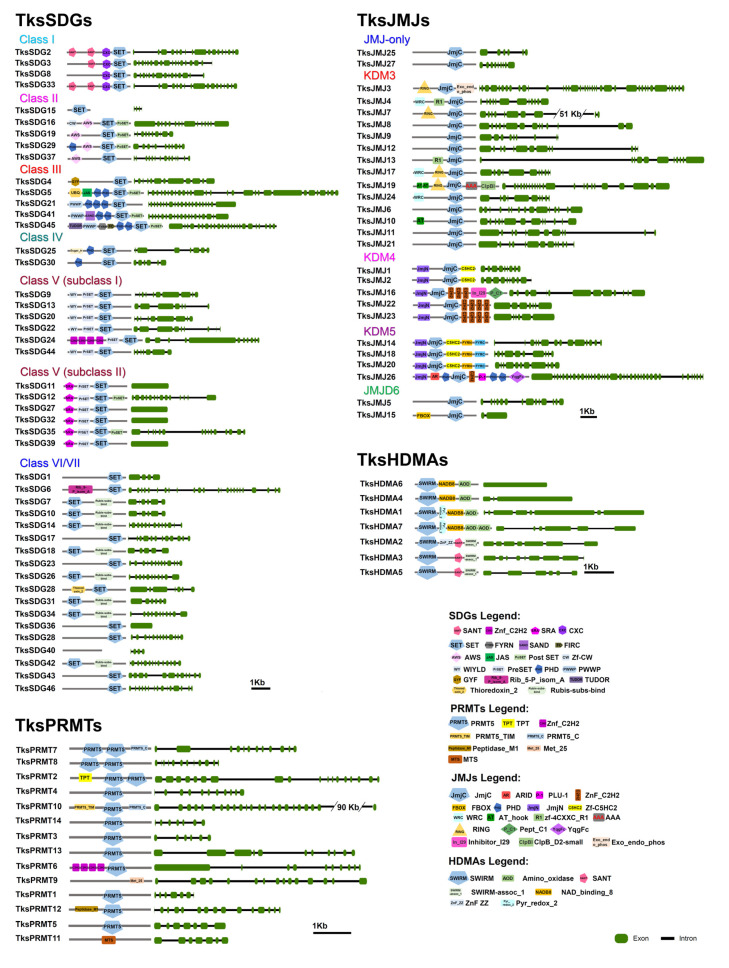
Domain composition and intron-exon structure of *Taraxacum kok-saghyz* HMTs and HDMs.

**Figure 3 plants-11-02077-f003:**
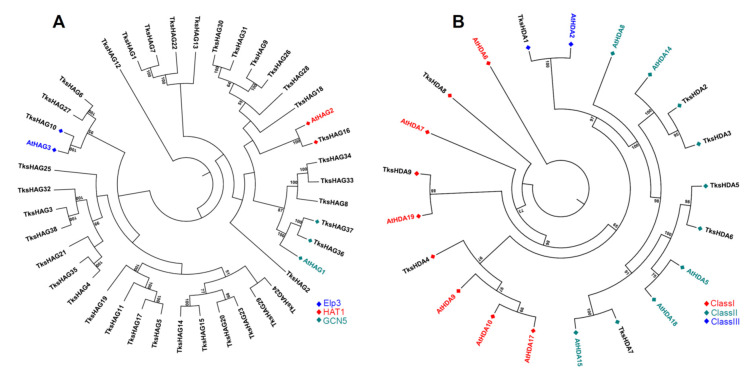
Phylogenetic tree of HAG (**A**) and HDA (**B**) proteins of *Taraxacum kok-saghyz* and *Arabidopsis thaliana* (in colour according to their respective classes). The numbers near the tree branches represent bootstrap values.

**Figure 4 plants-11-02077-f004:**
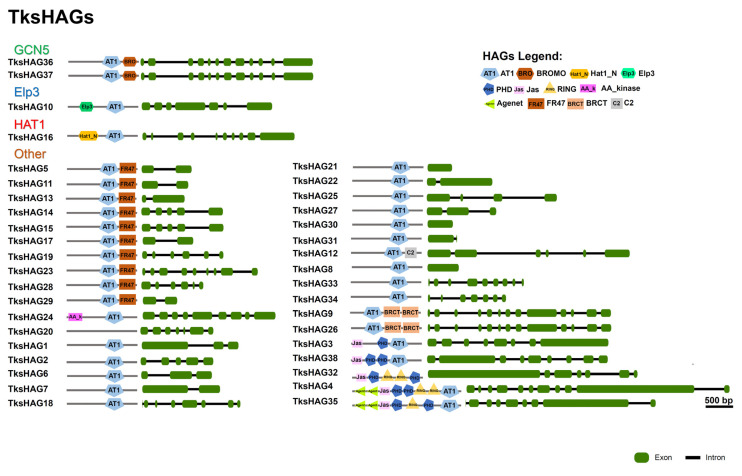
Domain composition and intron–exon structure of *Taraxacum kok-saghyz* HAGs.

**Figure 5 plants-11-02077-f005:**
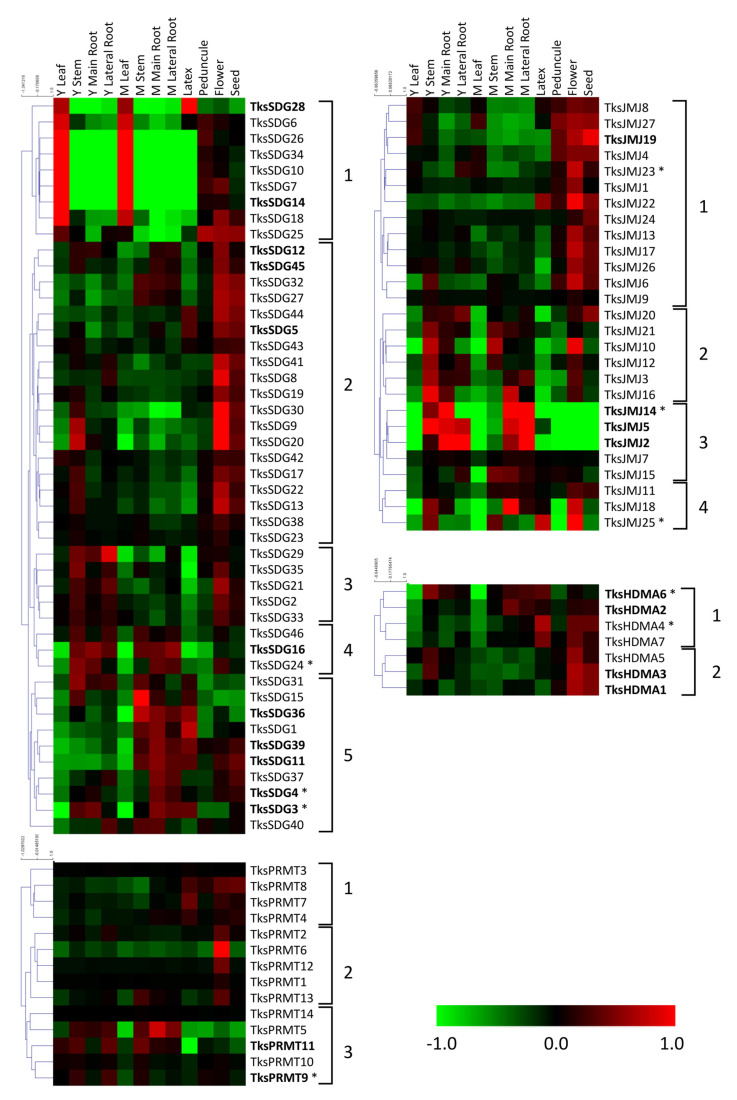
Heat map of *Tks*HMTs and *Tks*HDMs in different organs and developmental stages. The main clusters are indicated by square brackets. Genes with higher expression are indicated in bold. Asterisks indicate the genes analyzed by qPCR.

**Figure 6 plants-11-02077-f006:**
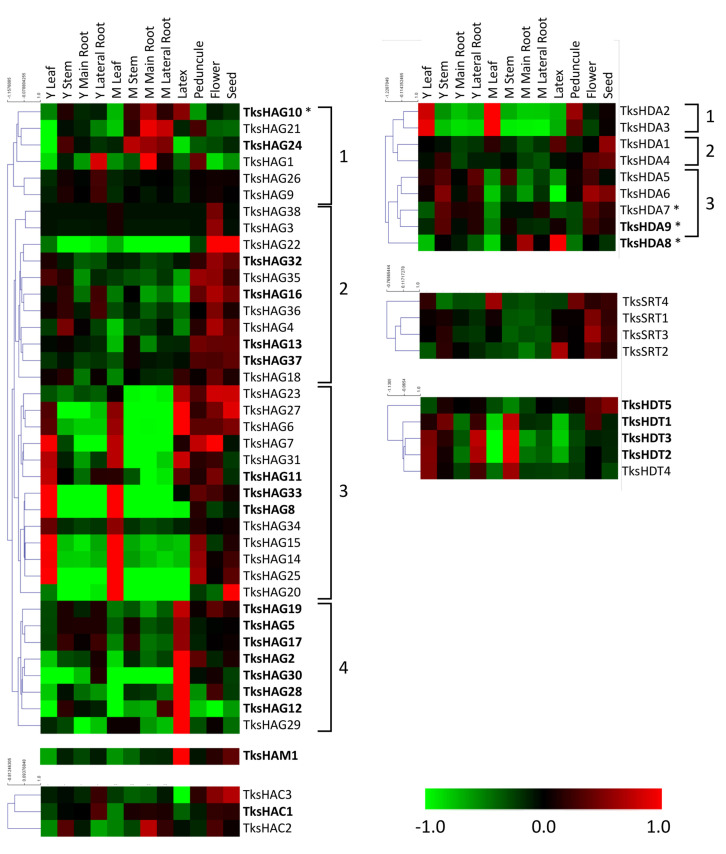
Heat map of *Tks*HATs and *Tks*HDACs in different organs and developmental stages. The main clusters are indicated by square brackets. Genes with a higher expression are indicated in bold. Asterisks indicate the genes analysed by qPCR.

**Figure 7 plants-11-02077-f007:**
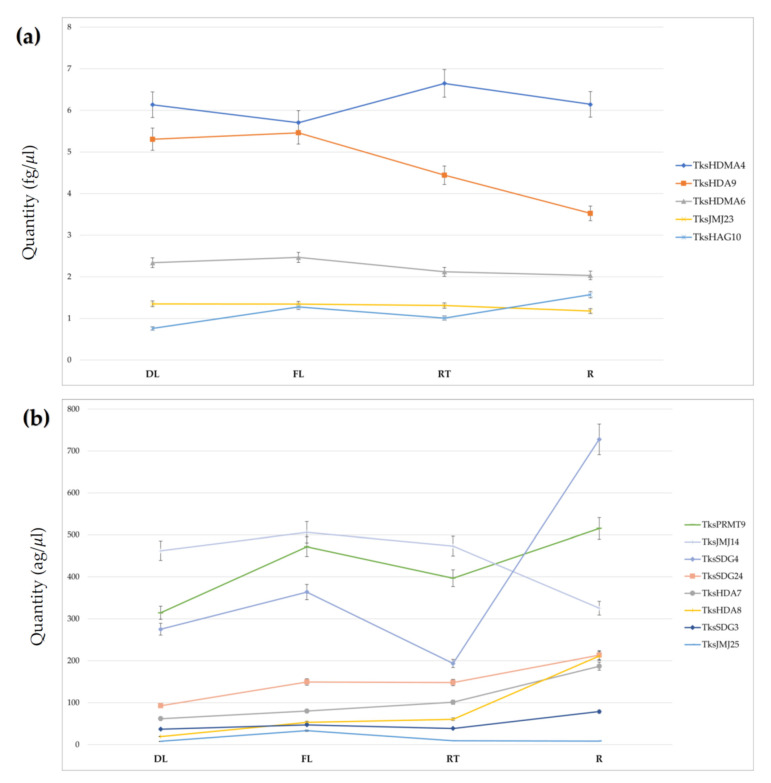
qPCR expression analysis of 13 HM genes in *Taraxacum kok-saghyz* developing leaves (DL), fully developed leaves (FL), root tips (RT), and mature roots (R). (**a**) Absolute quantification (mean ± SD fg/μL) of HM genes with an average expression above 1 fg/μL; (**b**) Absolute quantification (mean ± SD ag/μL) of HMG genes with an average expression below 1 fg/μL.

**Table 1 plants-11-02077-t001:** Number of genes identified in the different classes of HMGs in *Taraxacum kok-saghyz* and in three reference species: *Lactuca sativa*, *Arabidopsis thaliana*, and *Medicago truncatula*.

Family	Group	PFAM	Taraxacum Kok-Saghyz	Lactuca Sativa	Arabidopsis Thaliana	Medicago Truncatula v.4.01
HMT	SDG	PF00856	46	46	41	78
	PRMT	PF05185	14	4	7	3
HDM	JMJ	PF02373	27	47	21	34
	HDMA	PF04433	7	8	4	12
HAT	HAG	PF00583	38	34	3	51
	HAM	PF01853	1	2	2	1
	HAC	PF08214	3	14	5	11
	HAF	PF09247	0	1	2	1
HDAC	HDA	PF00850	9	19	12	10
	SRT	PF02146	4	3	2	2
	HDT		5	3	4	3
			**154**	**181**	**103**	**206**

## Data Availability

RNAseq reads archived as part of the PRJCA000437 bioproject were downloaded from https://ngdc.cncb.ac.cn/search/?dbId=gwh&q=%20PRJCA000437 (accessed at 15 May 2022) which is included in the CNCB/NGDC genome warehouse.

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
