# Peer review of "Genome-Wide Identification and Spatial Expression Analysis of Histone Modification Gene Families in the Rubber Dandelion Taraxacum kok-saghyz"

_plants, 2022, doi:10.3390/plants11162077_

Round 1

Reviewer 1 Report

Panara et al. Genome-Wide identification and spatial expression analysis of histone modification gene families in the rubber dandelion Taraxacum kok-saghyz

This is a nice cataloging of histone modification gene families in Taraxacum kok-saghyz. Yet it does not add much to our understanding of epigenetic control of cellular/nuclear processes in plants or to understanding epigenetic control of growth and development and response to stress in T. kok-saghyz. Neither how epigenetics regulates natural rubber (NR) production in the root of this plant.

The authors elaborated on the potential of the plant in producing NR and inulin and other metabolites for multiple purposes including bioethanol production and for uses in pharmacological and food industry. Since the authors have not conducted research addressing the (epigenetic) regulation of biosynthesis of these important substances they can shorten the introduction on this matter and provide a reference instead.

The authors concluded their Introduction section by stating: “Altogether, our work aims to define the epigenetic contribution towards natural rubber production in plants.”

It is difficult to see how the present work provides insight into the epigenetic contribution for NR production. This is a vague sentence and, in my view, should be omitted or rephrased.

Other than that, the bioinformatic/in silico analysis is fine. The only comment I have in this respect is on the categorization of HDT group as a group within histone deacetylases. There is no clear and direct experimental evidence for the function of members of the HDT group in Arabidopsis as histone deacetylases and many researchers have questioned their identity as HDACs. On the other hand, HD2-type HDACs share homology to a class of cistrans prolyl isomerases present in eukaryotes.

I suggest to discuss, in brief, this matter in the results section.

Minor comments:

Line 85, replace “is placed by HMTs” by ‘is catalyzed by HMTs’.

Line 86, should be HDM not HMD?

Line 86-89, The mode of action of HDMs is variable depending on the substrate and the type of JMJ. It has been reported that the mode of action could be through amine oxidation or deimination (methyl Argenine). JMJC-containing proteins act via hydroxylation of the methyl group within the methylated lysine residue. The references provided are not relevant here. Try this one: [Schneider J, Shilatifard A. Histone demethylation by hydroxylation: chemistry in action. ACS Chem Biol. 2006 Mar 17;1(2):75-81. doi: 10.1021/cb600030b. PMID: 17163647.]

Line 91, should be ‘dimethylated’.

Line 95, please add respectively after ‘tails’.

Table 1. HDMs, according to Lu et al., 2008 [Lu F, Li G, Cui X, Liu C, Wang XJ, Cao X (2008). Comparative analysis of JmjC domain-containing proteins reveals the potential histone demethylases in Arabidopsis and rice. J. Integr. Plant Biol. 50(7), 886–896.] Arabidopsis possesses 21 JMJ-C containing proteins.  

Lines 305-306. Please spell out ‘fg’ and ‘pg’ the first time they occur.

It will helpful adding statistical significance in Fig. 7.

In any case, the mRNA level does not necessarily reflect the protein level and it would be better complementing the qPCR analysis by either RNA seq or by proteome analysis of nuclear proteins.

“Figure S7. Phylogenetic tree of PRMT proteins of Taraxacum kok-saghyz and Arabidopsis (in bold). The numbers near the tree branches represent bootstrap values.”

The image indicates HDT rather than PRMT.

Reviewer 2 Report

1. Authors must perform two different bioinformatics analysis 1.  Synteny and 2. Ka/Ks ratio why Tk has more number of HMG 154 genes as compared with Arabidopsis HMG 101. They also need to check whether these genes tandem or segmental duplication.

2. Line no 77, SDGs are divided into 7 groups but explain only four. Authors need to check

3. Reference is missing in line no 120 for Arabidopsis

4. Fig no 2 needs to be rearranged and improve its quality so that it can become easily readable.  

5. It will be better if the authors provide all the gene information in the supplementary file that presented in table 1 

Round 2

Reviewer 1 Report

The authors have addressed most comments raised and the paper is suitable for publication.

Reviewer 2 Report

Authors have given satisfactory response.